# Childhood adversity and deliberate self-poisoning in Sri Lanka: a protocol for a hospital-based case–control study

Duleeka W Knipe,[1] Piumee Bandara,[2] Lalith Senarathna,[3] Judi Kidger,[1] José López-López,[1] Thilini Rajapakse[4]

DWK and PB are joint first authors.

[1]Population Health Sciences, Bristol Medical School, University of Bristol, Bristol, UK
[2]Translational Health Research Institute, Western Sydney University, Campbelltown, New South Wales, Australia
[3]Department of Health Promotion, Rajarata University of Sri Lanka, Mihintale, Sri Lanka
[4]Faculty of Medicine, University of Peradeniya, Peradeniya, Sri Lanka

**Correspondence to**
Dr Duleeka W Knipe;
dee.knipe@bristol.ac.uk

## ABSTRACT

**Introduction** Childhood adversity (CA) has been suggested as a key risk factor for suicidal behaviour, but evidence from low/middle-income countries is lacking. In Sri Lanka, CA, in the form of child maltreatment or as a consequence of maternal separation, has been highlighted in primarily qualitative or case series work, as a potentially important determinant of suicidal behaviour. To date, there have been no quantitative studies to investigate CA as a key exposure associated with suicidal behaviour in Sri Lanka. The aim of the research is to understand the association between CA and suicidal behaviour in Sri Lanka and to identify potentially modifiable factors to reduce any observed increased risk of suicidal behaviour associated with CA.

**Methods and analysis** This is a hospital-based case–control study. Cases (n=200) will be drawn from individuals admitted to the medical toxicology ward of the Teaching Hospital Peradeniya, Sri Lanka, for medical management of intentional self-poisoning. Sex and age frequency-matched controls (n=200) will be recruited from either patients or accompanying visitors presenting at the outpatient department and clinic of the same hospital for conditions unrelated to the outcome of interest. Conditional logistic regression will be used to investigate the association between CA and deliberate self-poisoning and whether the association is altered by other key factors including socioeconomic status, psychiatric morbidity, current experiences of domestic violence and social support.

**Ethics and dissemination** Ethics approval has been obtained from the Ethical Review Committee of the Faculty of Medicine, University of Peradeniya, Sri Lanka. Researchers have been trained in administering the questionnaire and a participant safety and distress protocol has been designed to guide researchers in ensuring participant safety and how to deal with a distressed participant. Results will be disseminated in local policy fora and peer-reviewed articles, local media, and national and international conferences.

## INTRODUCTION

Suicide is a significant cause of mortality resulting in approximately 800 000 deaths per year globally, 39% of which occur in the WHO region of South-East Asia.[1] WHO estimates suggest that cases of non-fatal self-harm

### Strengths and limitations of this study

- ► This is the first quantitative study to determine the association of childhood adversity and deliberate self-harm in Sri Lanka.
- ► This study will help identify potentially modifiable risk factors to inform the design of interventions and policies to reduce suicidal behaviour in the Sri Lankan context.
- ► Internationally validated instruments for the assessment of childhood adversity, depression, alcohol use and domestic violence and pretested forms for the assessment of other variables of interest are used.
- ► Hospital control outpatients may have a different exposure distribution compared with the base population—for example, they may have higher rates of mood disorders and suicidal ideation, introducing the potential for selection bias.
- ► The reported rate of abuse (childhood and current) may be underestimated due to cultural stigma and recall bias.

occur at least 20 times more frequently than the number of suicides.[2] In the last few decades, Sri Lanka, a middle-income country in South Asia, has experienced significant variations in its suicide rate, reaching its peak in 1995 of 47.0 per 100 000.[3 4] In response to the magnitude of this public health issue, a Presidential Task Force on Suicide Prevention was established in 1997, whose actions included decriminalising suicidal acts and restricting the availability of pesticides, the most common means of suicide in Sri Lanka.[5] Although reported rates of suicide have declined over the past two decades, studies of trends in intentional self-harm suggest that incidence of non-fatal intentional self-harm is increasing, especially among young people.[6 7] A better understanding of the risk factors for suicide and self-harm in these settings is essential for developing informed suicide prevention policies.

Childhood adversity (CA) has been suggested as a key risk factor for suicidal

behaviour,[8] but evidence from low/middle-income countries (LMIC) is lacking. Adverse childhood experiences may comprise acts of physical, sexual and emotional abuse as well as emotional and physical neglect and witnessing domestic violence.[9 10] It has been estimated that 30% of the risk of mental disorders in LMICs is attributable to CA,[11] and evidence from high-income countries reports that CA is a significant risk factor for suicidal behaviour.[8 12] However, the challenges faced in LMICs are likely to be different, as parental mortality is much higher and the rise in temporary labour migration has resulted in millions of children growing up with one or both parents absent, disrupting parental ties and family structures.[13] In Sri Lanka, CA, in the form of child maltreatment (eg, domestic violence), or as a consequence of maternal separation, has been highlighted in primarily qualitative or case series work as a significant contributor for suicidal behaviour.[14 15] To date, there have been no quantitative studies to investigate CA as a key exposure associated with suicidal behaviour in Sri Lanka, nor how CA might be associated with suicidal behaviour in the presence of other established sociodemographic and psychiatric risk factors. Current epidemiological evidence is lacking and is urgently needed as Sri Lanka formulates its suicide prevention activities.

The aim of this research is to investigate the association between CA and suicidal behaviour in Sri Lanka, and to identify potentially modifiable factors to reduce any increased risk of suicidal behaviour associated with CA (see figure 1—hypothesised conceptual model of self-harm). The results from this study will help inform the design of appropriate interventions to reduce the number of preventable suicide deaths in this country and thereby also reduce the number of individuals bereaved by suicide.

## METHODS AND ANALYSIS
### Study setting
The study will be carried out in the Teaching Hospital Peradeniya (THP), located in the Kandy District, Central Province of Sri Lanka. Kandy District has a total population of 1 375 382 and is 115 km east from Colombo, the capital of Sri Lanka.[16] The THP is a tertiary referral hospital with a catchment area that includes the North Central, North Western and Sabaragamuwa Provinces.

### Patient and public involvement
Priority of the research question, choice of outcome measures and questionnaire design were informed by discussions with community members. At the end of a previous study exploring the association between socioeconomic position and suicidal behaviour in Sri Lanka,[17] 10 community workshops were conducted in a rural area with a high risk of suicide. The workshops were conducted with community members aged 18 years and over, representing both sexes. A wide range of stakeholders were engaged, including: teachers, social workers, Grama Niladhari (village officers), community leaders/ members, national government officials (Ministry of Health, Education, Social and Welfare), non-governmental agencies, researchers and charities.[18] During the workshops community members engaged in discussions on the possible pathways to suicidal behaviour. In

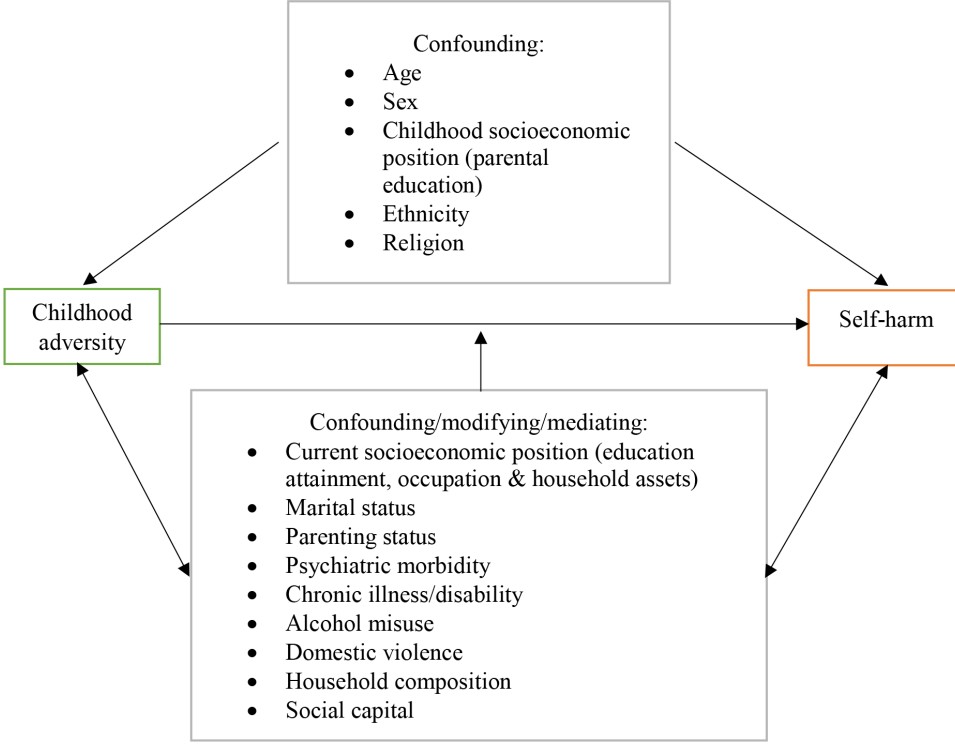

**Figure 1** Hypothesised conceptual model of self-harm.

all 10 workshops, child abuse, maltreatment and neglect were highlighted as important risk factors on the causal pathway and in the intergenerational cycle of poverty and heightened suicide risk. These discussions informed the design of the questionnaire and planned analysis in terms of factors considered as exposures and confounders. The questionnaire used in this study was also piloted with patients in the toxicology unit and outpatient department of the THP and revised as a result.

## Study design

An individually matched hospital-based case–control design will be used in this study. Cases will be drawn from individuals admitted to the medical toxicology ward (ward 17) of the THP (Sri Lanka) for medical management of deliberate self-poisoning. All persons presenting to the THP due to poisoning (accidental or deliberate) for emergency care are admitted to the toxicology unit for observation and treatment as needed. This case definition excludes self-harm due to other methods, for example, cutting; however, self-poisoning represents the most common method of self-harm cases presenting to hospital in Sri Lanka.[19–21]

Sex and age frequency-matched controls will be drawn from patients and accompanying visitors presenting to the outpatient department of the THP. Outpatients typically present with conditions such as cough, chest infection, hypertension, pregnancy and diabetes-related complications, conditions unrelated to the outcome of interest for this study.[22] Visitors and patients attending the nearby outpatient clinic for specialist care treatment and management will also be recruited.

## Inclusion and exclusion criteria

All patients aged 18 years and over, admitted to the study site during the 6-month study period, for medical management of deliberate self-poisoning, will be considered for inclusion in the study as a case. Patients and visitors attending the outpatient department and specialist clinics of the THP will be considered for recruitment as controls. All participants will be provided with the participant information sheet and will sign a consent form prior to being enrolled into the study.

Patients who are physically unable or too unwell to participate in the study interview prior to discharge from hospital, and those already diagnosed with an intellectual disability or dementia, will be excluded from the study. Those admitted for management of accidental poisoning will not be recruited. Accidental poisoning will be initially ascertained from the patient admission record and verbally reconfirmed by the patient through self-report.

As the focus of this study is on current suicidal behaviour, controls with a previous self-harm episode will not be excluded from the study but this information will be recorded. For every control with a previous self-harm episode, another control matched by sex and age with no previous self-harm will be recruited; and cases with a history of self-harm can be excluded in sensitivity analyses.

## Sample size

The estimated monthly case admission for intentional self-poisoning within ward 17 is approximately 100 cases per month. With the aim of collecting 50% of cases per month, we plan to recruit 200 cases in total and 200 age and sex frequency-matched controls over a 6-month time period. Assuming 20% of controls report a history of CA (odds=0.25), a twofold difference in risk will be detectable with 86% power (alpha=0.05).[8 11]

## DATA COLLECTION

The study is expected to recruit participants over a 6-month period from 18 July 2018 to 10 January 2019. Interviews will be conducted by trained data collectors in the participant's preferred language (Sinhala, Tamil or English). Interviewers will not be blinded to the case or control status of the participant and the same interviewers who recruit cases will also recruit controls. In order to minimise interviewer bias the interviewers will be given a standard script which they are requested to follow regardless of case status. The supervisor (PB) will regularly shadow interviewers to ensure that the interviewers adhere to the script.

Given the nature of the physical layout of the ward and outpatient department, and the sensitivity of the questions, interviews will only be conducted with the participant if they are able to accompany the data collector to a designated confidential space nearby, for interview. This ensures that the interview can be conducted in private and in the absence of accompanying friends or family members. Interviews will be conducted during non-visiting hours for patients in ward 17 to ensure responses will not be influenced by another person and for patient safety.

## Questionnaires
### Main exposure

The main exposure of interest in this study is CA. CA data will be collected using the WHO Adverse Childhood Experiences International Questionnaire, a culturally adapted screening tool for CA.[23] The questionnaire is designed for administration to people over the age of 18. Questions cover family dysfunction; physical, sexual and emotional abuse and neglect by parents or caregivers; peer violence; and witnessing community violence, and exposure to collective violence. The English questionnaire has been translated, back-translated and previously piloted into the two local languages.

In addition to the questions included in the WHO questionnaire, participants will be asked pretested questions about their experiences of parental (mother/father) absence due to temporary foreign migration during the first 18 years of life.

### Confounders and other study factors

Demographic data will be collected using a pretested questionnaire. Data on age, sex, ethnicity, religion,

marital status, residential area, household composition and parenting status (ie, presence and number of children and age of youngest child) will be collected. Employment status will also be ascertained, and a description of participant occupation will be defined based on the following categories: 'elementary occupation', 'armed forces', 'craft or related trades worker', 'skilled agricultural worker', 'service worker', 'technician', 'small business holder', 'big business holder', 'professional' and 'manager/legislator/administration'.

For socioeconomic position, data will be collected on individual educational attainment, parental education and motorised vehicle ownership (a proxy measure of household socioeconomic position). For individual and parental educational attainment, participants will be asked to select the highest completed level of education from the following categories: 'no schooling', 'completed between grades 1–5', 'completed between grades 6–10', 'passed ordinary level', 'passed advanced level' and 'completed university/postgraduate qualifications'. For motorised vehicle ownership, participants will be asked if they own a motorbike, three-wheeler vehicle, car/van, tractor and/or a bus.

Information on the type of poison ingested for the current self-harm episode will be collected based on the following categories: 'medicinal overdose', 'pesticide ingestion', 'plant poison', 'petroleum-based products', 'rat poison', 'other household chemical' or 'other'. Where possible, participants will be asked to specify the name of the poison ingested. Suicidal intention and lethality of the attempt will not be assessed due to constraints on the length of the questionnaire.

Data on past self-harm behaviour will be collected via self-report. Participants will be asked if they have ever previously self-harmed. This will be recorded regardless of whether or not the episode resulted in hospital presentation. Participants will also be asked if they know of a close friend or family member who has self-harmed or died by suicide during the past year. Past psychiatric morbidity (ie, diagnosed with a mental disorder) and whether or not the participant experiences existing comorbidities, such as chronic illness and physical disabilities, will also be collected.

Data on current psychiatric morbidity will also be collected. Participants will be asked to complete the 9-item Patient Health Questionnaire (PHQ-9). The PHQ-9 is a brief, one-page, self-administered questionnaire that is internationally validated for assessing the severity of depression. The PHQ-9 has been translated and validated for use in the Sinhala-speaking Sri Lankan population.[24] The translated questionnaire has previously been used in the National Mental Health Survey of Sri Lanka for both Sinhala and Tamil populations.[25]

Information on alcohol use disorders will be collected, based on the Alcohol Use Disorders Identification Test (AUDIT), which has been developed by the WHO and has also been validated for the local population,[26] and has been previously used in Sri Lanka. The AUDIT comprises questions related to the frequency and quantity of alcohol consumption and the effect of alcohol consumption on behaviour.

Participants will also be asked if they have learnt about sexual and reproductive health rights through school and if the information delivered was useful. In addition, data on current exposure to domestic violence will be collected using a translated, back-translated and piloted version of the four-question Humiliation, Afraid, Rape, Kick (HARK) questionnaire.[27] The four short questions aim to capture different components of domestic violence, relating to emotional abuse, psychological abuse, sexual violence and physical violence.

Finally, participants will also be asked questions relating to their social capital, including their emotional support networks and sense of belonging. The questions used in this survey are questions that have been used as part of a large social capital community survey in the North Central Province of Sri Lanka.[28] Questions are designed to capture whether participants are currently emotionally supported at the household and community levels and if they feel a sense of belonging. Information will also be collected on past social capital based on a sense of belonging at school, and interaction with peers and teachers.[29 30]

## Analysis plan

To ensure that questionnaires are as complete as possible, the supervisor (PB) will review data missingness on a regular basis to ensure that data collectors are not consistently missing information. Once the data collection has been finalised, the level of missingness will be assessed. It is anticipated that any missingness will not be missing at random (a requirement for imputation) and therefore missing data will not be imputed in the main analyses. Instead, it is anticipated that our main analyses will be based on complete cases only, excluding case–control pairs that contain missing data. A full case analysis (regardless of missing) will be conducted to explore whether excluding case–control pairs with missing data might have introduced bias in the results.

Conditional logistic regression techniques will be used to examine the association between CA and deliberate self-poisoning (outcome). In the primary analysis, CA will be quantified using the total score of the WHO Adverse Childhood Experiences International Questionnaire, and unadjusted associations with the outcome will be presented. Additionally, separate analyses will be conducted using questions about parental absence (a proxy for CA) and including both the WHO questionnaire and the questions about parental absence as separate covariates in the model to explore the partial association of each with the outcome.

The study will investigate the potential confounding effects of parental education, ethnicity and religion in a series of models. Potential confounders will be specified as covariates in models with measures of CA and parental absence as the main exposures of interest,

following investigation of univariate associations between each potential confounder and self-harm. Additional key factors, such as current socioeconomic status, marital and parental status, psychiatric morbidity, current experiences of domestic violence and social support, will also be incorporated to investigate potential effect measure changes. The statistical software Stata V.15[31] will be used for data analysis.

## ETHICS

Each participant will be given a verbal explanation of the study with a written information sheet (in their native language) and they will be given time to read it. Permission to recruit will be sought via written informed consent. Participants will be informed during their consent process that the interview is voluntary and that they have the right to withdraw at any time during or after the interview. Participants will also be informed about the purpose of the study, the members of the research team, the reason they have been chosen for the study, consequences of participation (potential benefits and disadvantages), confidentiality, potential outcomes of the research and contact details of the principal investigator for further information. If the researcher suspects that the participant does not have the cognitive functioning to give informed consent, the individual will not be recruited for the study.

The study includes questions related to self-harm, CA and current domestic violence which may result in participant distress. A participant safety and distress protocol has been designed to guide researchers in ensuring participant safety and how to deal with a distressed participant. With regard to risk of self-harm, the protocol advises that if the participant reports experiencing suicidal thoughts daily during the preceding two weeks, they will be referred to the Psychiatry Clinic, THP, held every Thursday for further management and follow-up. Where project staff have reason to believe that the participant is at an immediate risk to themselves or others, the psychiatric on-call doctor will be notified. Participants are informed that they are eligible to withdraw from the study at any point, and that this will have no adverse effect on their medical or psychiatric management or follow-up. All interviews will be conducted in a manner that ensures that the participants' usual medical treatment or care does not get delayed or adversely impacted due to study participation.

If during the completion of this questionnaire the participant discloses a child safeguarding issue (ie, current sexual abuse of a child, under the age of 18), the project staff will notify the principal investigator (DWK), and the local principal investigator (TR) will inform the National Child Protection Authority, as appropriate. Participants will be informed during the consent process that such disclosures will result in notification. The circumstances under which confidentiality would be broken will be explained to the participant during the informed consent process.

Participants will be asked questions about domestic abuse, and if these questions are asked in the presence of the perpetrator of that abuse or other family members this may have adverse outcomes for the participant. In recognition of this risk, the interview will only be conducted in private during non-visiting hours and in a private location in the hospital. If current domestic abuse is disclosed, the participant will be offered information about help available locally and if appropriate, the patient will also be offered a referral to the psychiatry clinic.

## DISSEMINATION

Findings from this research will be published academically in peer-reviewed articles and will also be given publicity in the local media. Findings may also be disseminated at national and international conferences, health symposia and local policy fora, for example, local government and non-government agencies. Any participant who is interested can also receive a copy of the report of the research and the outcomes from the investigators.

**Acknowledgements** The authors thank the senior academics who have acted as advisors for the study: Professor David Gunnell, Professor Gene Feder, Professor Chris Metcalfe (University of Bristol), Professor Michael Eddleston (University of Edinburgh) and Professor Flemming Konradsen (University of Copenhagen). The authors also thank the community members who informed the direction of the research and study design. The authors thank Professor Andrew Page (Western Sydney University) for his valuable feedback on the manuscript and the staff at SACTRC, in particular Indunil Abeyratne and Chamil Kumara, for their support in setting up the study. In addition, the authors acknowledge the significant contribution of the data collection team Kasuni Silva, Tharuka Silva, Sandareka Samarakoon and Azra Aroos, and the staff on ward 17 and the OPD of the Teaching Hospital Peradeniya for accommodating this research.

**Contributors** DWK, LS, JK, JLL and TR were responsible for study concept, design and funding acquisition. DWK and TR wrote the protocol. PB drafted the manuscript and coordinated the manuscript preparation and revision. DWK, TR and PB were responsible for piloting the survey, and PB was responsible for supervising the data collection. All authors provided critical evaluation and revision of the manuscript and have given final approval of the manuscript accepting responsibility for all aspects.

**Funding** This work is supported by the UK Medical Research Council (grant number MC_PC_MR/R019622/1), and the Elizabeth Blackwell Institute for Health Research, University of Bristol and the Wellcome Trust Institutional Strategic Support Fund.

**Competing interests** None declared.

**Patient consent for publication** Not required.

**Ethics approval** Ethics approval was granted from the Ethical Review Committee of the Faculty of Medicine, University of Peradeniya, Sri Lanka, on 14 June 2018.

**Provenance and peer review** Not commissioned; externally peer reviewed.

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
