## [Reviewer comments · BMJ Open]

ARTICLE DETAILS

TITLE (PROVISIONAL)	Childhood adversity and deliberate self-poisoning in Sri Lanka: a protocol for a hospital-based case-control study
AUTHORS	Knipe, Duleeka; Bandara, Piamee; Senarathna, Lalith; Kidger, Judi; López-López, José; Rajapakse, Thilini

VERSION 1 – REVIEW

REVIEWER	Rory O'Connor University of Glasgow Scotland
REVIEW RETURNED	05-Feb-2019

GENERAL COMMENTS	This paper reports the protocol for an ambitious case control study to investigate the relationship between childhood adversity and self-poisoning in Sri Lanka The study has the potential to make a very useful contribution to the research literature. My comments are minor and are limited to seeking some more information about the study (and I appreciate that the study design has already been approved by ethics). 1. I may have missed it, but I couldn't find the start and end dates. Please add in.2. Please add more information on the consenting process.3. More detail on the matching process would be helpful. E.g., will this be conducted by same people who recruit 'cases' and what procedures are in place to minimise bias?4. If I understand the design, the focus is on those who are admitted to the hospital, rather than those who are discharged following presentation to emergency care? Please clarify.5. How will 'accidental' self-poisoning be ascertained? Similarly how will past self-harm in controls be determined? Records, self-report? Does previous episode mean self-harm which required hospitalisation or any self-harm?6. Can authors add a brief section about how they will deal with missing data?7. Given the debate around suicidal versus non-suicidal self-harm, did the authors consider recording suicidal intent?8. Analysis, ethics and dissemination are all appropriate. I wish the authors well in their study recruitment.
--

REVIEWER	Mauri Marttunen, professor University of Helsinki, Finland
REVIEW RETURNED	01-Mar-2019

GENERAL COMMENTS	The ms is a study protocol of a study on childhood adversity (CA) as a risk factor for suicidal behavior in Sri Lanka. The study is important as little is known on the association of CA and suicidal behaviour in low and middle-income countries. I have some concerns detailed below.  1. According to the BMJ Open instructions for study protocols the dates of the study should be included in the manuscript 2. Cases will be drawn from individuals admitted to a medical toxicology ward for medical management of intentional self-poisoning. Why is suicidal behaviour limited to deliberate self-poisoning? Why not include also other forms of suicide attempts with suicidal intention? Why are suicidal intention and lethality of the attempt not assessed? 3. Controls with a previous self-harm episode will not be excluded from the study but this information will be recorded. What is the rationale for this decision? These controls are likely to have same kind of risk factors (including CA) as the patients admitted to the medical toxicology ward. 4. Psychiatric disorders are common among people who make serious suicide attempters. Therefore, assessment of the attempters' psychiatric morbidity is important. According to the study protocol, the researchers are planning to assess current psychiatric morbidity only with two self-report scales, PHQ-9 (depression) and AUDIT (alcohol use). The authors should give a rationale for  a) Measuring only depressive symptoms and alcohol use. Why are they not planning to assess also other psychopathology? b) Using only self-report measures in assessing psychopathology. Why are they not planning to use diagnostic interview? Although the reviewer is not an expert in statistics the analysis plan seems to be accurate. The researchers have paid much attention to ethical issues. I find the chapter on ethics exemplary.
--

REVIEWER	Roger Mulder University of Otago, Christchurch New Zealand
REVIEW RETURNED	10-Mar-2019

GENERAL COMMENTS	Comments to the authors The study is a Protocol for a hospital-based case-control study on childhood adversity and deliberate self-poisoning in Sri Lanka. As the authors note the study is particularly important because it takes place in a low and middle income country where there is very limited data on suicidal behavior. They also note that it is important because childhood adversity in such countries may be different from
--

	that in high income countries. This is particularly related to parental mortality and temporary labor migration. The study is a collaboration between Sri Lankan, UK and Australian researchers. The methods and analysis are well described and the sample size appropriate. The control groups in such studies are always a problem. I presume selecting visitors and patients from attending the nearby outpatient clinic is down to convenience. This is reasonable as it stands but I think the authors need to note potential confounders around using such a population. There is some evidence that people with medical conditions attending outpatient clinics have higher rates of mood disorders and possibly suicidal ideation than the general population. Obviously it would be better to get randomly selected controls but I can understand this would be very difficult. Nevertheless the authors may need to consider their recruitment of controls in the limitations section. It is a strength that the interviews can be conducted in the participants preferred language and using a standard script. While it would be better if they were blinded to the patient's status it would obviously be very difficult practically to do this. The questionnaires appear appropriate and have been carefully translated and used in previous Sri Lankan studies. The analysis plan is also appropriate. One can always argue about potential confounders but these ones chosen seem reasonable in this situation. The ethics have been carefully considered and dissemination appropriate.
--	---

VERSION 1 – AUTHOR RESPONSE

Reviewer #1:

“This paper reports the protocol for an ambitious case control study to investigate the relationship between childhood adversity and self-poisoning in Sri Lanka.

The study has the potential to make a very useful contribution to the research literature. My comments are minor and are limited to seeking some more information about the study (and I appreciate that the study design has already been approved by ethics).”

We thank the Reviewer for their comments.

I may have missed it, but I couldn't find the start and end dates. Please add in.

We thank the Reviewer for their observation. We have incorporated the start and end dates in the revised manuscript under 'Data Collection' (page 5, paragraph 1).

The study is expected to recruit participants over a 6-month period, from 18 July 2018 to 10 January 2019.

Please add more information on the consenting process.

We have included further information on the consenting process in the revised manuscript under 'Ethics' (page 7, paragraph 3).

Each participant will be given a verbal explanation of the study with a written information sheet (in their native language) and they will be given time to read it. Permission to recruit will be sought via written informed consent. Participants will be informed during the consent process that the interview is voluntary and that they have the right to withdraw at any time during or after the interview.

Participants will also be informed about the purpose of the study, the members of the research team, the reason they have been chosen for the study, consequences of participation (potential benefits and disadvantages), confidentiality, potential outcomes of the research, and contact details of the Principal Investigator for further information. If the researcher suspects that the participant does not have the cognitive functioning to give informed consent, the individual will not be recruited for the study.

More detail on the matching process would be helpful. E.g., will this be conducted by same people who recruit 'cases' and what procedures are in place to minimise bias?

The same interviewers who recruit cases will also recruit controls. This statement has been added to the revised manuscript under 'Data collection' (page 5, paragraph 1). We note that we have acknowledged how interview bias will be minimised in the original manuscript under 'Data Collection' (page 5, paragraph 1).

Interviewers are not blinded to the case or control status of the participant and the same interviewers who recruit cases will also recruit controls. In order to limit any minimise interviewer bias, the interviewers will be given a standard script which they are requested to follow regardless of case status. The supervisor (PB) will regularly shadow interviewers to ensure that the interviewers adhered to the script.

If I understand the design, the focus is on those who are admitted to the hospital, rather than those who are discharged following presentation to emergency care? Please clarify.

Yes, cases will be those admitted to the medical toxicology ward for ongoing medical management of deliberate self-poisoning as described in the 'Study design' (page 3, paragraph 5) of the original manuscript.

To clarify, all persons presenting to the Teaching Hospital Peradeniya (THP) due to poisoning (accidental or deliberate) for emergency care are admitted to the toxicology unit (ward 17) of THP for observation and treatment as needed. They are not managed and discharged directly from the emergency treatment unit (ETU), which only keeps patients for a maximum of 4 hours.

We have added this information under 'Study design' (page 4, paragraph 2).

An individually matched hospital-based case-control design will be used in this study. Cases will be drawn from individuals admitted to the medical toxicology ward (ward 17) of the THP (Sri Lanka) for medical management of deliberate self-poisoning. All persons presenting to the THP due to poisoning (accidental or deliberate) for emergency care are admitted to the toxicology unit (ward 17) for observation and treatment as needed.

How will 'accidental' self-poisoning be ascertained? Similarly how will past self-harm in controls be determined? Records, self-report? Does previous episode mean self-harm which required hospitalisation or any self-harm?

We have added a statement on how accidental self-harm will be ascertained under 'Inclusion and exclusion criteria' (page 4, paragraph 5).

Those admitted for management of accidental poisoning will not be recruited. Accidental poisoning will be initially ascertained from the patient admission record and verbally reconfirmed by the patient through self-report.

Any previous self-harm, regardless of whether or not the episode resulted in hospital presentation, will be recorded by self-report. This has been incorporated in the revised manuscript under the 'Confounders and other study factors' section (page 6, paragraph 1).

Data on past self-harm behaviour will be collected via self-report. Participants will be asked if they have ever previously self-harmed. This will be recorded regardless of whether or not the episode resulted in hospital presentation. Participants will also be asked if they know of a close friend or family member who has self-harmed or died by suicide during the past year.

Can authors add a brief section about how they will deal with missing data?

We have added a section on how we will deal with missing data under the 'Analysis plan' (page 6, paragraph 6).

To ensure that questionnaires are as complete as possible, the supervisor (PB) will review data missingness on a regular basis to ensure that data collectors are not consistently missing information. Once the data collection has been finalised, the level of missingness will be assessed. It is anticipated that any missingness will not be missing at random (a requirement for imputation) and therefore missing data will not be imputed in the main analyses. Instead, it is anticipated that our main analyses will be based on complete cases only, excluding case-control pairs that contain missing data. A full case analysis (regardless of missing) will be conducted to explore whether excluding case-control pairs with missing data might have introduced bias in the results.

Given the debate around suicidal versus non-suicidal self-harm, did the authors consider recording suicidal intent?

Suicidal intention and lethality of the attempt will not be assessed due to constraints on the length of the questionnaire. To minimise interview burden and ensure the WHO Adverse Childhood Experience International Questionnaire (the main exposure of interest) could be administered in its entirety we made the decision to exclude the assessment of suicidal intention. We also considered it important to assess for common psychiatric co-morbidity, such as depression and alcohol use disorders. Therefore, we chose to focus on these assessments.

A brief statement has been added to the revised manuscript under 'Confounders and other study factors' (page 6, paragraph 3).

Suicidal intention and lethality of the attempt will not be assessed due to constraints on the length of the questionnaire.

Analysis, ethics and dissemination are all appropriate. I wish the authors well in their study recruitment.

We thank the Reviewer for their time in considering this manuscript and for their comments.

Reviewer #2:

The ms is a study protocol of a study on childhood adversity (CA) as a risk factor for suicidal behavior in Sri Lanka. The study is important as little is known on the association of CA and suicidal behaviour in low and middle-income countries.

We thank the Reviewer for their comments.

According to the BMJ Open instructions for study protocols the dates of the study should be included in the manuscript

We thank the Reviewer for their observation. We have incorporated the start and end dates in the revised manuscript under 'Data Collection' (page 5, paragraph 1).

The study is expected to recruit participants over a 6-month period, from 18 July 2018 to 10 January 2019.

Cases will be drawn from individuals admitted to a medical toxicology ward for medical management of intentional self-poisoning. Why is suicidal behaviour limited to deliberate self-poisoning? Why not include also other forms of suicide attempts with suicidal intention? Why are suicidal intention and lethality of the attempt not assessed?

Self-poisoning represents the most common method of deliberate self-harm cases presenting to hospital in Sri Lanka as referenced in the 'Study design' (page 3, paragraph 5). While other deliberate self-harm methods are used (e.g. cutting), these are difficult to capture at the Teaching Hospital Peradeniya and it was beyond the scope and resources of the current study to recruit non-self-poisoning cases.

Suicidal intention and lethality of the attempt were not assessed due to constraints on the length of the questionnaire. To minimise interview burden and ensure the WHO Adverse Childhood Experience International Questionnaire (the main exposure of interest) could be administered in its entirety, we made the decision to exclude the assessment of suicidal intention. We also considered it important to assess for common psychiatric co-morbidity, such as depression and alcohol use disorders. Therefore, we chose to focus on these assessments.

A brief statement has been added to the revised manuscript (page 6, paragraph 3).

Suicidal intention and lethality of the attempt will not be assessed due to constraints on the length of the questionnaire.

Controls with a previous self-harm episode will not be excluded from the study but this information will be recorded. What is the rationale for this decision? These controls are likely to have same kind of risk factors (including CA) as the patients admitted to the medical toxicology ward.

In our control selection we didn't exclude previous self-harm, as the focus of the study is current and not past behaviour. However, for every control with a previous self-harm episode, another control matched by sex and age with no previous self-harm will be recruited; and cases with a previous history of self-harm can be excluded in sensitivity analyses. Our preliminary analysis reveals that 0.05% of controls had a previous self-harm attempt.

We have incorporated further detail in the revised manuscript under 'Inclusion and exclusion criteria' (page 4, paragraph 6).

As the focus of this study is on current suicidal behaviour, controls with a previous self-harm episode will not be excluded from the study but this information will be recorded. For every control with a previous self-harm episode, another control matched by sex and age with no previous self-harm will be recruited; and cases with a previous history of self-harm can be excluded in sensitivity analyses.

Psychiatric disorders are common among people who make serious suicide attempters. Therefore, assessment of the attempters' psychiatric morbidity is important. According to the study protocol, the researchers are planning to assess current psychiatric morbidity only with two self-report scales, PHQ-9 (depression) and AUDIT (alcohol use). The authors should give a rationale for

a) Measuring only depressive symptoms and alcohol use. Why are they not planning to assess also other psychopathology?

b) Using only self-report measures in assessing psychopathology. Why are they not planning to use diagnostic interview?

The rationale for a) and b) were logistical.

a) Other psychopathologies were not assessed due to constraints on the length of the questionnaire. To minimise interview burden and ensure the WHO Adverse Childhood Experience International Questionnaire (the main exposure of interest) could be administered in its entirety, we made the decision to limit the assessment of psychiatric morbidity to depression and alcohol use using scales that have been adapted and validated for use in Sri Lanka.^{1,2} Furthermore, evidence from a systematic review (currently under review) of 112 studies from low and middle-income countries shows that the prevalence of psychiatric disorders was 47% among those who self-harmed, with the most common psychiatric disorder being depression.³

b) The resources/time required for a clinical diagnostic interview was not feasible given the sample size and study setting.

Although the reviewer is not an expert in statistics the analysis plan seems to be accurate.

The researchers have paid much attention to ethical issues. I find the chapter on ethics exemplary.

We thank the Reviewer for their time in considering this manuscript and for their comments.

Reviewer #3:

The study is a Protocol for a hospital-based case-control study on childhood adversity and deliberate self-poisoning in Sri Lanka.

As the authors note the study is particularly important because it takes place in a low and middle income country where there is very limited data on suicidal behavior. They also note that it is important because childhood adversity in such countries may be different from that in high income countries. This is particularly related to parental mortality and temporary labor migration. The study is a collaboration between Sri Lankan, UK and Australian researchers.

We thank the Reviewer for their comments.

The methods and analysis are well described and the sample size appropriate. The control groups in such studies are always a problem. I presume selecting visitors and patients from attending the nearby outpatient clinic is down to convenience. This is reasonable as it stands but I think the authors need to note potential confounders around using such a population. There is some evidence that people with medical conditions attending outpatient clinics have higher rates of mood disorders and possibly suicidal ideation than the general population. Obviously it would be better to get randomly selected controls but I can understand this would be very difficult. Nevertheless the authors may need to consider their recruitment of controls in the limitations section.

We acknowledge that the recruitment of controls from the nearby outpatient department is down to convenience and accept the Reviewer's comment that the recruitment of controls from the outpatient department will need to be considered in the limitations section. We have added further detail in the 'Strengths and Limitations' section of the revised manuscript (page 2).

Hospital control outpatients may have a different exposure distribution compared to the base-population – for example they may have higher rates of mood disorders and suicidal ideation, introducing the potential for selection bias.

It is a strength that the interviews can be conducted in the participants preferred language and using a standard script. While it would be better if they were blinded to the patient's status it would obviously be very difficult practically to do this.

We have acknowledged in the original manuscript that interviewers will not be blinded and note how we will minimise interviewer bias. This is noted in the 'Data Collection' section (page 5, paragraph 1).

Interviewers are not blinded to the case or control status of the participant and the same interviewers who recruit cases will also recruit controls. In order to limit any minimise interviewer bias, the interviewers will be given a standard script which they are requested to follow regardless of case status. The supervisor (PB) will regularly shadow interviewers to ensure that the interviewers adhered to the script.

The questionnaires appear appropriate and have been carefully translated and used in previous Sri Lankan studies. The analysis plan is also appropriate. One can always argue about potential confounders but these ones chosen seem reasonable in this situation. The ethics have been carefully considered and dissemination appropriate.

References

Hanwella R, Ekanayake S, de Silva VA. The Validity and Reliability of the Sinhala Translation of the Patient Health Questionnaire (PHQ-9) and PHQ-2 Screener. *Depress Res Treat* 2014; 2014: 768978.
Hanwella R, de Silva VA, Jayasekera NE. Alcohol use in a military population deployed in combat areas: a cross sectional study. *Subst Abuse Treat Prev Policy* 2012; 7: 24.
Knipe D, Williams AJ, Hannam-Swain S, Upton S, Brown K, Bandara P, Chang S, Kapur N. Psychiatric morbidity and suicidal behaviour in low and middle-income countries: A systematic review and meta-analysis. Manuscript submitted for publication 2019.